# Poisoning-Induced Acute Kidney Injury: A Review

**DOI:** 10.3390/medicina60081302

**Published:** 2024-08-12

**Authors:** Ching-Hsiang Yu, Lan-Chi Huang, Yu-Jang Su

**Affiliations:** 1Department of Emergency Medicine, MacKay Memorial Hospital, Taipei 10449, Taiwan; allenyu1895@gmail.com; 2Department of Emergency Medicine, MacKay Memorial Hospital, Tamshui Branch, New Taipei City 251020, Taiwan; flanchih@gmail.com; 3Toxicology Division, Department of Emergency Medicine, MacKay Memorial Hospital, Taipei 10449, Taiwan; 4Department of Medicine, MacKay Medical College, New Taipei City 25245, Taiwan; 5MacKay Junior College of Medicine, Nursing, and Management, Taipei 11260, Taiwan; 6Department of Nursing, Yuanpei University of Medical Technology, Hsinchu 300, Taiwan

**Keywords:** acute kidney injury (AKI), rhabdomyolysis, lithium, heroin, herbicide

## Abstract

Acute kidney injury (AKI) is a debilitating, multi-etiological disease that is commonly seen in clinical practice and in the emergency department. In this review, we introduce the definition, symptoms, and causes of poisoning-related AKI; we also discuss its mechanisms, risk factors, and epidemiology, as well as elaborate on the relevant laboratory tests. Subsequently, we discuss the treatment strategies for toxin- and substance-related AKI caused by Glafenin, antimicrobial agents, lithium, contrast media, snake venom, herbicides, ethylene glycol, synthetic cannabinoids, cocaine, heroin, and amphetamines. Finally, for a comprehensive overview of poisoning-related AKI, we review the management, prevention, and outcomes of this condition.

## 1. Acute Kidney Injury (AKI): Definition, Incidence, and Symptoms

Acute kidney injury (AKI) is characterized by a duration of ≤7 days and is defined by functional criteria, including a ≥50% rise in serum creatinine (sCr) within 7 days, a ≥0.3 mg/dL (26.5 µmol/L) rise in sCr within 2 days, or oliguria persisting for ≥6 h [1]. Contemporary classifications such as the Risk, Injury, Failure, Loss of kidney function, and End-stage kidney disease (RIFLE) and Acute Kidney Injury Network (AKIN) systems offer updated definitions, encompassing a spectrum from minor sCr elevations to the necessity of renal replacement therapy (RRT) [2]. The 2012 Kidney Disease: Improving Global Outcomes (KDIGO) guidelines further refine the diagnostic criteria, eliminating the need for adequate fluid resuscitation or the exclusion of urinary obstruction before an AKI diagnosis [3]. Patients with chronic kidney disease (CKD) face an augmented risk of AKI, rendering diagnosis challenging due to variability in baseline kidney function. The presence of AKI in CKD patients poses substantial risks, with AKI stage 3 defined by an increase in serum creatinine to >4 mg/dL [3,4]. In the RIFLE criteria, acute kidney injury is defined as ↑ SCr × 2 or ↓ GFR >50% compared to the baseline [2].

The global burden of AKI-related problems exceeds that of breast cancer, diabetes, or heart failure, with high mortality rates persisting over the past 50 years [5]. Incidence patterns vary between community-acquired and hospital-acquired AKI, with higher-income countries predominantly reporting hospital-acquired cases and lower-income settings experiencing more community-acquired cases [6]. Mortality rates range from 23% to 49.4%, with contributing factors including surgical or diagnostic interventions, sepsis, volume depletion, and toxin exposure [7]. The imperative for enhanced innovation in AKI prevention and treatment is underscored by its impact on one in five hospitalized patients globally (20.0 to 31.7% of patients at various levels of hospitalization), calling for dialog among healthcare professionals, policymakers, and the public to raise awareness and develop improved hospital-based healthcare delivery systems focusing on AKI prevention, early detection, and treatment [8].

AKI often lacks the immediate, alarming symptoms, like chest pain or dyspnea, seen in other types of acute organ failure. However, in advanced-stage AKI (stage 3), disturbances in the acid–base balance, electrolyte levels, and uremic toxin accumulation may lead to symptoms such as tachypnea, metabolic acidosis, and an increased bleeding risk [4,9].

Other non-specific symptoms include weakness, malaise, fatigue, and decreased appetite. Notably, urine discoloration can offer valuable insights into the etiology of AKI, particularly in rhabdomyolysis, where muscle cell injury leads to the excessive release of muscle enzymes into the serum, leading to myoglobinuria and AKI. Major signs include pain, weakness, and swelling of the affected muscles, a reddish discoloration of the urine, and oligo-anuria [10]. Various urine colors may signify different conditions; for instance, green urine may indicate a Pseudomonas-related urinary tract infection, black urine could suggest paraphenylenediamine dye intoxication, red urine may be associated with scrub typhus-related disseminated intravascular coagulation, and orange–brown urine may be observed in obstructive jaundice patients receiving multivitamins [11]. Therefore, recognizing these urine color changes can facilitate prompt AKI diagnosis, particularly when associated with rhabdomyolysis, infection, drugs, or toxins.

## 2. AKI in the Emergency Department (ED): Risk Factors, Causes, and Etiology

AKI presents a significant challenge in emergency department (ED) settings, with diverse risk factors contributing to its occurrence. Studies conducted across different regions have identified key risk factors associated with AKI presentations in the ED. Among these, hemodynamic instability emerges as a significant risk factor, with shock, including hypovolemic, cardiogenic, and septic shock, being a major contributor to prerenal AKI cases [12]. Additionally, comorbid conditions such as prior stroke or transient ischemic attack (TIA), dementia, heart failure, hypertension, coronary artery disease, or chronic kidney disease (CKD) have been consistently linked to an increased risk of AKI presentation in the ED [13]. Furthermore, demographic factors such as advanced age and the male gender have also been identified as risk factors for AKI in the ED population. For instance, in a retrospective cohort study conducted in Vancouver, Canada, older age and the male gender were associated with a higher prevalence of AKI among ED patients [14].

The etiology of AKI in the ED is multifactorial, encompassing a spectrum of prerenal, renal, and postrenal causes. Among the cases identified in a study in Tehran, Iran, 690 (89.61%) were classified as prerenal or renal, with prerenal causes accounting for 74 (73.3%) cases due to various types of shock, while renal causes, such as rhabdomyolysis, medication-induced nephrotoxicity, and chemotherapy-induced nephropathy, contributed to 35.0%, 17.5%, and 15.3% of cases, respectively [15]. Hypotension and renal vascular insufficiency represent the predominant prerenal etiologies, with hypovolemic shock and renal artery stenosis being common contributors [15]. The renal causes of AKI include conditions such as rhabdomyolysis, medication-induced nephrotoxicity, and chemotherapy-induced nephropathy, highlighting the diverse range of insults affecting renal function. Postrenal causes such as obstructive uropathies, including kidney stones, can also lead to AKI [15]. Notably, poisoning- and drug-induced AKI—accounting for nearly 20% of acute AKI episodes, with older adults being particularly vulnerable—stand out as crucial etiologies that require special attention in the ED setting due to their potentially severe consequences [16].

AKI presentations in the ED are influenced by a combination of risk factors and etiologic factors. Hemodynamic instability and comorbid conditions significantly contribute to AKI risk in the ED population. Understanding the diverse etiologies of AKI, including poisoning- and drug-induced causes, is essential for its prompt recognition and appropriate management in the ED. Collaborative efforts involving emergency physicians, pharmacists, and real-time AKI analysis may aid in mitigating the burden of AKI-related morbidity and mortality in the ED setting. Therefore, it is imperative for ED healthcare providers to remain vigilant and proactive in identifying and managing AKI cases to optimize patient outcomes and reduce the burden on healthcare resources.

## 3. AKI in the Emergency Department (ED): Epidemiology

AKI is a significant concern in emergency departments (EDs) globally, accounting for a notable proportion of patient presentations. In a comprehensive examination of two urban EDs in Vancouver, Canada, involving 1651 consecutive patients, the prevalence of AKI was determined to be 5.5% [14]. Similarly, across a retrospective cohort study spanning five EDs in the United States, 10.4% of cases met the KDIGO criteria for any stage of AKI upon ED arrival [17]. Moreover, a cross-sectional investigation conducted in Tehran, Iran, shed light on the prevalence of AKI in the ED, estimated at 315 cases per 1,000,000 population [15], underscoring the considerable burden it imposes in this clinical setting.

An analysis of gender distribution among AKI patients in the ED reveals a consistent trend toward male predominance across various studies. For instance, in the Vancouver study, 63% of AKI patients identified as male [14]. Similarly, data from Tehran, Iran, indicated that males comprised 59.1% of the AKI cohort [15]. In the United States, a higher proportion of males were observed to present with community-acquired AKI compared to females (52.8%) [17]. These findings emphasize the importance of considering demographic variables in the comprehension of AKI epidemiology.

With respect to age distribution, AKI patients in the ED exhibit a wide age range. In Tehran, Iran, the mean age of AKI patients was reported to be 62.72 years, with 61.9% of individuals aged 60 years or older [15]. Contrastingly, in the United States, older individuals were more likely to present with community-acquired AKI, with a median age of 66 years for AKI cases compared to 57 years for non-AKI cases [17]. Similarly, in Vancouver, AKI patients were found to be a median of 23 years older than non-AKI patients, highlighting the correlation between age and AKI risk in the ED setting [14]. These observations underscore the necessity for tailored, age-specific strategies in both the management and prevention of AKI among ED patients.

In summary, AKI poses a substantial burden in the ED, with prevalence rates ranging from 5.5% to 10.4% across studies conducted in Vancouver, Canada, the United States, and Tehran, Iran [14,15,17]. The consistent male predominance among AKI patients underscores the significance of demographic factors in understanding AKI epidemiology. Furthermore, the wide age range of AKI patients emphasizes the need for age-specific approaches in AKI management and prevention within the ED setting. These findings collectively underscore the urgency of addressing AKI as a critical issue in EDs and advocate for tailored interventions to mitigate its multifaceted impact.

## 4. Poisoning-Related AKI: Mechanisms, Etiology, and Laboratory Investigations

Acute kidney injury (AKI) resulting from poisoning involves a complex interaction of mechanisms targeting renal tubular cells, rendering them susceptible to toxic insults [16,18]. Certain medications and toxins, such as aminoglycosides, contrast agents, and vancomycin, exert their harmful effects by disrupting mitochondrial function within the renal tubules [19]. The presence of “muddy brown casts” in the urine is pathognomonic for acute tubular necrosis (ATN), a common consequence of such toxic insults [16]. Additionally, acute interstitial nephritis, often triggered by a hypersensitivity reaction to medications like beta-lactams, quinolones, and non-steroidal anti-inflammatory drugs, underscores the non-dose-related inflammatory injury characteristic of this condition [20]. Factors precipitating crystal-induced AKI include underlying kidney disease, volume depletion, and metabolic disturbances favoring changes in urinary pH conducive to crystal formation. Glomerular injury, a manifestation of nephritic syndrome, can result from various medications such as bisphosphonates and hydralazine, further complicating the etiological landscape of poisoning-related AKI [21].

The clinical manifestations of poisoning-related AKI encompass a broad spectrum of symptoms, often reflecting the underlying toxic insult. In developing countries, self-poisoning with pesticides and herbicides predominates, whereas in developed nations, antibiotic toxicity, non-steroidal anti-inflammatory drugs, and contrast media are frequently implicated [22]. In cases of poisoning-related AKI, the clinical presentation may vary widely, ranging from dysuria and hematuria to oligo-anuria and systemic arterial hypertension [23]. Notably, illicit drugs, including synthetic cannabinoids, can precipitate AKI alongside other systemic complications, necessitating comprehensive evaluation and supportive care [24].

Laboratory investigations play a pivotal role in the diagnosis and management of poisoning-related AKI, offering insights into the underlying pathophysiology and guiding therapeutic interventions. Urinalysis, a cornerstone of AKI evaluation, may reveal characteristic findings such as proteinuria, hematuria, and the presence of specific crystals indicative of the underlying toxic insult [16]. Toxicological analysis of blood and urine samples aids in identifying the culprit toxins and informing targeted treatment strategies. Furthermore, biochemical markers such as serum creatinine and electrolyte levels provide valuable prognostic information and help monitor renal function during treatment [25].

Urine color abnormalities, ranging from red to black, offer valuable diagnostic clues and underscore the diverse etiology of poisoning-related AKI [26]. Red urine, a notable consequence of hydroxocobalamin administration for cyanide poisoning, serves as a distinctive marker of treatment [27]. Conversely, brown urine, often mistaken for red urine due to its darker hue, may signal underlying disorders or ingestions such as acetaminophen overdose, highlighting the importance of thorough evaluation [28]. Black urine, while less common, can result from various medications, including metronidazole, nitrofurantoin, and methocarbamol, necessitating a comprehensive workup to discern potential systemic toxicity [29,30]. Blue and green urine discoloration, attributed to methylene blue ingestion or exposure to certain pesticides and herbicides, underscores the diverse range of substances capable of altering urine color [31,32]. Purple urine, a rare manifestation associated with purple urine bag syndrome, implicates Gram-negative bacteriuria and resolves with appropriate antibiotic therapy and catheter management [33]. Additionally, white urine may indicate severe urinary tract infection or urinary tuberculosis, warranting meticulous urine culture and antibiotic therapy for resolution [34]. Overall, urine color abnormalities serve as crucial indicators of potential poisoning-related AKI and necessitate prompt evaluation and management to mitigate adverse outcomes.

## 5. Poisoning- or Drug-Induced AKI

Not every substance or toxin leads to AKI: its occurrence depends on the toxin’s dynamic, and some affect the lungs or liver rather than the kidneys. For a systemic overview of poisoning- and drug-induced AKI, we searched the literature in the PubMed database, limiting the search to human studies, using the keywords “acute poisoning” and “intoxication”. The results we obtained are listed in Table 1 and include the publication year, substance, country, additional remarks, and references.

## 6. Treatment Strategies for Toxin- and Substance-Induced AKI

The mechanism of AKI varies depending on the substance or toxin that has led to it, so, in this section, we discuss available treatment strategies according to the causative agent.

**Glafenin** is a non-steroidal anti-inflammatory drug (NSAID), and its hydrochloride salt is used for the treatment of all types of pain. Due to a high incidence of anaphylactic reactions, there are concerns about its safety and its withdrawal from the market is currently being considered. Glafenin can also cause interstitial tubular nephritis and AKI [35]. Therefore, the use of NSAIDs should be avoided in people with eGFR <30 mL/min per 1.73 m^2^, and prolonged use should be avoided in those with eGFR 30–59 mL/min per 1.73 m^2^ [1]. **Vancomycin** is a potent, time-dependent antimicrobial agent, and its nephrotoxicity can vary from mild in some patients to AKI deterioration and a need for RRT initiation in others. This is avoidable, but the dose needs to be regulated and used according to the patient’s renal function (glomerular filtration rate, GFR). The other bactericidal agents of note here are **aminoglycosides**, which are wide-spectrum, concentration-dependent antimicrobials effective against Gram-negative bacteria. The incidence of aminoglycoside-induced AKI is approximately 33%, and it occurs due to the presence of poor perfusion in critically ill elderly patients. In a study conducted in Israel, AKI incidence was as low as 5.6% and was not associated with amikacin administration, even in those with chronic renal impairment or in those who had progressed to AKI on admission [59]. **Gentamicin** can influence glomerular filtration due to mesangial contraction and cell proliferation. Increased vascular resistance and a decrease in renal blood flow have also been noted. In using these kinds of nephrotoxic antibiotics, including aminoglycosides, the physician must carefully monitor the therapeutic drugs used to prevent nephrotoxic serum concentrations and the development of AKI [60].

**Lithium** is commonly used in the treatment of bipolar affective disorder; however, 20% of patients are affected by its nephrotoxic properties, exerted through the mechanism of chronic tubulointerstitial nephropathy, leading to interstitial fibrosis, thickening of the tubular basement membranes, and renal tubular cysts. In an Australian report published in 2024, a total of 437 patients with lithium-related kidney failure were compared to 1280 controls with kidney failure not related to lithium usage from the Australian and New Zealand Dialysis (ANZDATA) registry. Patients with lithium-associated kidney failure received RRT at a significantly older age (by 4 years, 62 vs. 58, *p* < 0.001) and were more likely to be female (63% vs. 40%, *p* < 0.001), European (93% vs. 68%, *p* < 0.001), and live in a higher-socioeconomic-status (SES) (*p* < 0.001) district. Despite this, there were no differences in survival [61].

**Contrast**-induced acute kidney injury (CI-AKI) has become one of the major causes of hospital-acquired AKI, which is a serious concern among patients. Patients who receive iodinated contrast medium face a nearly 5.5% risk of AKI occurrence [60]. The precise pathogenesis of CI-AKI may be related to its direct cytotoxicity, hypoxic ischemia of the medulla, or to oxidative stress caused by high-osmolality iodinated radiographic media, which have diverse physicochemical properties, including cytotoxicity, permeability, and viscosity. Contrast medium-induced nephropathy is defined as iatrogenic disease occurring after the intravascular injection of a high-osmolality iodinated radiographic medium with a rise in serum creatinine >25% or a 0.5 mg/dL (44 µmol/L) absolute increase over the baseline at least 1–3 days after contrast administration. Generally, CI-AKI can be prevented by hydration, antagonistic vasoconstriction, and antioxidant drugs [62]. There is no consensus on the suitable speed, volume, and regimen of hydration to prevent AKI; however, the general principles are diluting the concentration of the contrast medium, reducing its viscosity, accelerating its excretion, and reducing its retention time. Among other medications, N-acetylcysteine (NAC), a direct scavenger of free radicals, is thought to have a protective effect against CI-AKI due to it reducing reactive oxygen species (ROS) production, improving blood flow, and dilating blood vessels through NO-mediated pathways [62].

**Snakebite** is not uncommonly seen in daily emergency department (ED) practice. Kidney injury after snakebite envenomation is not uncommon, with a reported prevalence of up to 32% [63]. Hypnale, or the hump-nosed viper, is a very venomous snake found in South Asia, e.g., India and Sri Lanka. In an observation study of hump-nosed viper-bitten patients (*n* = 465), 9.5% of cases involved AKI; of these, 14% of AKI patients died of renal complications and coagulopathy. The most common clinical presentations of envenomation are hematuria, oliguria, microangiopathic hemolysis, thrombotic thrombocytopenic purpura, and hemolytic uremic syndrome. Antivenom must be administered immediately in these patients to save lives [60].

**Hymenoptera** stings are also commonly seen in daily emergency practice. Bees and wasps are widespread all over the world, and their stings typically only cause mild erythematous and edematous local reactions, needing anti-inflammatory agents, antihistamines, and the topical application of antibiotic ointment. In some, however, Ig-E-mediated anaphylactic shock may occur, which, if followed by cardiovascular or multi-organ complications, including progressing to AKI, can be life-threatening. An especially hypersensitive reaction can lead to rhabdomyolysis, with skeletal muscle cell damage and subsequent myoglobinuria and AKI [60]. There are some risk factors for AKI after wasp stings: these include the number of stings, aspartate aminotransferase (AST) higher than 147 U/L, lactate dehydrogenase higher than 477 U/L, time from sting to admission above 12 h, and activated partial thromboplastin time >49 s [64]. AKI cases after wasp stings have markedly elevated IL-6, IL-10, and Il-17 levels compared to non-AKI patients [65].

As for **herbicides**, **glyphosate** is a weak acidic solution that can lead to significant necrotic and apoptotic damage to proximal tubular cells and thickened ascending limb of the loop of Henlé in the outer medulla and cortex. One study found that the incidence of **glyphosate**-induced AKI was 44.5%, of which 25% of patients were classified as presenting a risk of AKI, 13% as having renal failure, and 6.5% as affected by renal injury [66]. The renal toxicity of **paraquat** manifests itself 24 to 96 h after exposure and the histopathological findings of animal studies on **paraquat** poisoning are dose-dependent, with proximal and distal tubular cell lesions showing acute tubular necrosis [60]. An Indian report from 2013 noted that, unusually, AKI had developed in all documented cases of **paraquat** poisoning (100%), and the associated mortality rate was 66% [67].

**Ethylene glycol** intoxication is associated with calcium oxalate deposits in the urine, resulting in severe AKI. The histological findings are acute tubular necrosis, cytoplasmic vacuolization, and refractile calcium oxalate crystals in proximal tubules [60]. **Ethylene glycol** toxicity can induce an elevated anion gap, metabolic acidosis, central nervous system dysfunction, cardiovascular compromise, and AKI. One study noted that individuals more likely to face devastating effects or death were older, male, and presented with more severe symptoms requiring intensive care, with a mortality rate of 0.3% [68].

As **new psychoactive substances (NPSs)**, most **synthetic cannabinoids** (SCBs) are more potent and efficacious at affecting CBRs than traditional THC. It is possible for SCBs, especially given the various combinations found in SCB products, to be mixed with other NPSs. The pharmacodynamics of synthetic cannabinoids can potentially differ in various physiological and psychological disturbances, with effects that are probably mediated by the actions of SCBs at non-cannabinoid receptors. Microscopically, renal biopsy in people with SCB poisoning and acute renal failure reveals acute tubular damage and acute tubulointerstitial nephritis. The active components of marijuana and SCBs are likely metabolized differently and thus have distinct pharmacokinetic profiles; the active metabolites of SCBs likely contribute their effects by activating cannabinoid receptors (CBRs) [24]. Renal damage is followed by elevated serum creatinine levels in 89% of patients [69].

**Cocaine**-induced AKI is a consequence of several pathological processes and has serious complications, such as rhabdomyolysis, with the destruction of skeletal muscle cells due to acute interstitial nephritis, ischemia, and vasoconstriction, renal infarction due to vasoconstriction and thrombosis, thrombotic microangiopathy, and malignant hypertension originating from endothelial injury and thrombocyte activation [60]. The acute insult can be via rhabdomyolysis, thrombotic microangiopathy, vasculitis, acute interstitial nephritis (AIN), and kidney infarction. Myocyte breakdown further worsens rhabdomyolysis [70].

In **heroin** abusers, viral disease-induced nephropathy is mostly associated with concomitant chronic hepatitis C and human immunodeficiency virus (HIV) infection, with secondary focal segmental glomerulosclerosis (FSGS) or membranoproliferative and membranous glomerulonephritis (MPGN) and a clinical presentation of nephrotic syndrome. In several cohorts, the seroprevalence rates for HCV, HBV, and HIV were 85, 81, and 24 to 26%, respectively [71]. The presence of concomitant infectious processes, such as HIV infection, in heroin users may contribute to the development of renal injury.

In these patients, kidney biopsy confirms acute tubular interstitial impairment with inflammatory infiltration and edema [60]. Acute kidney injury (AKI) is a serious and sometimes fatal complication of rhabdomyolysis, and it occurs in 8 to 20% of rhabdomyolysis incidents [72].

As for commonly used illicit drugs, such as **amphetamine**, methamphetamine, and 3,4 methylenedioxymethamphetamine (MDMA, ecstasy), metabolized on the apical membrane of renal proximal tubular cells, the extracellular event of redox cycling appears to be a possible pathophysiological pathway of MDMA nephrotoxicity [60]. In one study, complications included rhabdomyolysis in 30% of cases and progression to acute kidney injury in 13% [55].

## 7. Management of Poisoning-Related AKI

Therapeutic measures in cases of acute intoxication include the following: gastrointestinal decontamination, ideally performed within one hour; specific antidote administration in specific intoxication, if available, as a priority; aggressive support care; and general empiric therapeutic approaches in all poisoned patients [60,73]. Activated charcoal, synthetic treatments, or anion exchange resins are available for the treatment of specific poisonings. It is only metals and corrosive substances that activated charcoal does not work on. Additionally, the molecular adsorbent recirculating system is a type of albumin dialysis that can be used in patients intoxicated with Amanita phalloides and in those with hepatorenal failure [60].

General recommendations for the treatment of patients with AKI are early identification, treatment, and investigation, especially in patients with mental and substance use disorders. Reducing the exposure to chemical agents is a necessary step for the recovery of renal function. Other essential steps include monitoring the patient’s vital signs and reducing the absorption of substances, improving excretion, correcting hypovolemia by using isotonic crystalloids while also avoiding hyperhydration, using inotropic agents to maintain a mean arterial pressure of 65 to 70 mmHg in hypotensive cases, using diuretic agents to avoid fluid overload-related pulmonary edema, monitoring therapeutic drug levels when initiating therapy with nephron-fragile antimicrobials (e.g., vancomycin, gentamicin), and applying RRT, correlated to the patient’s clinical status or laboratory findings if necessary [22,60,74]. Moreover, kidney sonography should be performed to exclude anatomical or structural anomalies.

Emergent RRT is necessary for AKI patients with an acid–base imbalance, hyperkalemia (K^+^ higher than 6.5 mmol/L), severe metabolic acidosis (pH less than 7.1), severe azotemia causing uremic symptoms, or fluid overload, which may lead to lung edema, dyspnea, and electrolyte imbalance [22,60]. RRT can be used in two forms, according to the duration and intensity needed: one is intermittent and the other is continuous hemodialysis, hemofiltration, or hemodiafiltration. Hemodynamic instability is usually a crucial factor for initiating continuous RRT [60]. The clinical and biochemical criteria to initiate RRT depend on many parameters, including the toxic substance (whether it is dialyzable or not), metabolic disorders, fluid volume overload or lack thereof, and severity of renal impairment. Typically, dialyzable substances include alcohol, salicylate, theophylline, lithium, vancomycin, barbiturate, metformin, carbamazepine, phenytoin, valproic acid, aminoglycosides, ethylene glycol, methanol, and ethanol [60,61,75,76]. In hepatorenal syndrome or certain kinds of poisoning, such as bongkrekic acid, acute hepatic failure as a complication of multi-organ failure results in a high mortality rate, reaching 90%. Albumin dialysis allows for the partial replacement of some of the liver’s excretory functions: the molecular absorbent recirculating system (MARS) can buy the patient precious time and bridge them to transplantation or hepatic regeneration [58,77].

For physicians in general ICUs or EDs, it is essential to use plasma exchange and blood purification in patients with intoxication- or poisoning-induced AKI. For example, hemoperfusion on adsorbent columns can be utilized when the toxin is lipophilic and has a high molecular weight. RRT is an important life-saving procedure which is widely used by specialized clinical staff with expertise in intensive care [58,60].

## 8. Outcomes and Prevention of AKI in the Emergency Department

Not every AKI patient can return to the baseline renal status that they had before the insult. The severity and duration of acute kidney injury influence both its short- and long-term outcomes. Despite recent definitions, only a few studies have assessed the pattern of renal recovery, and time-dependent competing risks have usually been overlooked. In one report, the cumulative incidence of AKI recovery was 25% on day 2 and 35% on day 7; recovery pattern classification was only achieved in 75% of critically ill patients and the remaining 25% needed long-term hemodialysis therapy [78]. Comprehensive knowledge of and training in nephrotoxic substances causing AKI will improve clinical practice in all fields of medicine. Complete patient histories, including exposure time, amount, route of contact, and time to presentation at the emergency department—as well as whether management was immediate or not—is the key to achieving better outcomes. If the causative substance is unknown, recognizing the toxidrome is necessary, as it may even allow for the use of an antidote according to the presenting symptoms, thus reducing the harm to renal function and ultimately preventing AKI [61].

## Figures and Tables

**Table 1 medicina-60-01302-t001:** Lists of acute poisoning or intoxication-related acute kidney injury are limited in human studies.

Year	Substance	Country	Additional Remarks	Reference
1981	Glafenin	Belgium	18 months’ duration	[35]
1989	Lithium	Denmark	0.8% incidence of AKI	[36]
1989	Cocaine (“crack”)	United States	Recovery from AKI within 96 h	[37]
1996	Ethylene glycol	Sweden	5.6% incidence of AKI	[38]
1998	Amanita proxima	France	83% incidence of AKI	[39]
2001	Acetaminophen	United States	8.9% incidence of AKI	[40]
2005	Diethylene glycol (DEG)	Argentina	51.7% incidence of AKI	[41]
2007	Roundup, a glyphosate-based herbicide	United States	Impaired glomerular filtration	[42]
2008	Methanol	Spain	Concurrent compartment syndrome	[43]
2009	Amantadine	Japan	Hyperkalemia	[44]
2009	Vitamin D and vitamin A intoxication	Brazil	Hypercalcemia	[45]
2009	Paraquat	Republic of Korea	Creatinine was higher in elevated pancreatic enzyme cases	[46]
2009	Acetaminophen	France	Acute tubular necrosis	[47]
2013	Ethylene glycol	France	Calcium oxalate in renal tubules can result in nephrocalcinosis and AKI	[48]
2015	Synthetic cannabinoid	New Zealand	Delayed seizure	[49]
2015	Mushroom	Italy	Orellanus syndrome	[50]
2015	Acetaminophen	Taiwan	2.4-fold increase in risk of AKI	[51]
2016	Vitamin D	India	Hypercalcemia	[52]
2018	Paraquat	Taiwan	Serum creatinine level improved gradually	[53]
2018	Heavy dose (40mg) colchicine	China	Treated with continuous renal replacement therapy (CRRT)	[54]
2019	Methamphetamine	Australia	13% incidence of AKI	[55]
2019	Pufferfish	United States	Tetrodotoxin poisoning	[56]
2019	Paraquat	China	38.2% cases of AKI happened in children	[57]
2024	Bongkrekic acid	Taiwan	Multi-organ failure	[58]

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
