# Peer review of "Poisoning-Induced Acute Kidney Injury: A Review"

_medicina, 2024, doi:10.3390/medicina60081302_

Round 1

Reviewer 1 Report

Comments and Suggestions for Authors

The authors have provided an extensive narrative review of AKI in general and drug- and poison-induced AKI in particular. I consider this review to be meaningful as it summarizes a lot of previous information. However, some modifications may be necessary to make the content more complete and easier to understand. Moreover, I recommend that the English language be proofread again by a native English speaker with expertise in the field.

Specific comments

Page 1

The most recent AKI guideline is the KDIGO 2012 guidelines, as indicated by the authors. The sudden mention of AKI stage 3 is not clear to the reader and should be explained more fully than the RIFLE criteria.

Page 1

The term kidney replacement therapy and renal replacement therapy appear in the text. If they refer to the same thing, the notation should be unified. Also, if the abbreviation is indicated, the abbreviation should be used, and there is no need to mention renal replacement therapy (RRT) every time. Authors should check the entire text and ensure consistency of expression.

Page 5-7 Strategy for variable toxin and substance in AKI

The narrative review in this section appears to be the most important in this paper. Some modifications are suggested to make it more meaningful.

Page 6 line 1

The suggestion to avoid dosing patients with Chronic renal insufficiency is confusing to the reader as there is no definition of renal insufficiency; it should be stated from which level of CKD stage and eGFR the risk increases as far as possible. I recommend that the phrase chronic renal insufficiency should not be used in the description.

Page 6 Lithium

This section only gave a general description of Lithium and epidemiological data. The mechanism of impairment and possible symptoms should be described, as is done for other drugs.

Page 6 Contrast

It is clear that complementary fluid therapy is effective against Contrast-induced AKI. It is helpful for the reader to clearly describe any such obvious preventive measures.

Page 6 Hymenoptera

The term “painkillers” used here does not seem appropriate as a medical term.

Page 7 heroin

The expression “vial disease-induced nephropathy” does not seem common and should be explained. It is unclear what is responsible for the kidney biopsy findings in the second half; if it is a direct effect of Heroin, it should be clearly stated.

Page 8 line34-36

Examples of lipophilic and high molecular weight toxins should be provided.

Author Response

Point-to-point replying to Reviewer 1

A: thanks for the valuable suggestions, and we revised it accordingly.

The authors have provided an extensive narrative review of AKI in general and drug- and poison-induced AKI in particular. I consider this review to be meaningful as it summarizes a lot of previous information. However, some modifications may be necessary to make the content more complete and easier to understand. Moreover, I recommend that the English language be proofread again by a native English speaker with expertise in the field.

A: We have completed the English editing service via https://www.mdpi.com/authors/english

Specific comments

Page 1

The most recent AKI guideline is the KDIGO 2012 guidelines, as indicated by the authors. The sudden mention of AKI stage 3 is not clear to the reader and should be explained more fully than the RIFLE criteria.

A: we just to introduce the AKI definition on CKD patients and also add information of RIFLE criteria. Patients with chronic kidney disease (CKD) face an augmented risk of AKI, rendering diagnosis challenging due to variability in baseline kidney function. The presence of AKI in CKD patients poses substantial risks, with AKI stage 3 defined by a serum creatinine increase to >4 mg/dl (3,4). In RIFLE criteria, acute kidney injury is defined as↑ SCr × 2 or ↓ GFR >50% of the baselines (2).

Page 1

The term kidney replacement therapy and renal replacement therapy appear in the text. If they refer to the same thing, the notation should be unified. Also, if the abbreviation is indicated, the abbreviation should be used, and there is no need to mention renal replacement therapy (RRT) every time. Authors should check the entire text and ensure consistency of expression.

A: Thanks for the excellent advice, and we unified as RRT.

Page 5-7 Strategy for variable toxin and substance in AKI

The narrative review in this section appears to be the most important in this paper. Some modifications are suggested to make it more meaningful.

Page 6 line 1

The suggestion to avoid dosing patients with Chronic renal insufficiency is confusing to the reader as there is no definition of renal insufficiency; it should be stated from which level of CKD stage and eGFR the risk increases as far as possible. I recommend that the phrase chronic renal insufficiency should not be used in the description.

A: We added the ‘So, it is avoided to use NSAIDs in people with eGFR <30 ml/min per 1.73 m2, and to avoid prolonged use in those with eGFR 30–59 ml/min per 1.73 m2 [1]’ by reference from KDIGO guideline.

To prevent confusion to readers, we omit the chronic renal insufficiency and change it to ‘ In using these kins of aminoglycoside and nephrotoxic antibiotics, the physician must use therapeutic drug monitoring to prevent nephrotoxic serum concentration and development of AKI’. 

Page 6 Lithium

This section only gave a general description of Lithium and epidemiological data. The mechanism of impairment and possible symptoms should be described, as is done for other drugs.

A: yes, we added information of ‘however, its nephrotoxic properties affected 20% of using patients by the mechanism of chronic tubulointerstitial nephropathy leading to interstitial fibrosis, thickening of the tubular basement membranes and renal tubular cysts.’

Page 6 Contrast

It is clear that complementary fluid therapy is effective against Contrast-induced AKI. It is helpful for the reader to clearly describe any such obvious preventive measures.

A: we omit the ‘currently, the effective therapy for CI-AKI is limited at present.’, and added ‘There is no consensus on the suitable speed, volume, and regimen of hydration, however, keeping the principles of dilution of contrast concentration, reduction of viscosity, acceleration of excretion, and reduction of retention time of contrast agents is the way to prevent AKI. Other medication such as N-acetylcysteine (NAC) was the direct scavenger of free radicals and is thought to have a protective effect on CI-AKI by reducing reactive oxygen species (ROS) production, improving blood flow, and dilating blood vessels through NO-mediated pathways.’

Page 6 Hymenoptera

The term “painkillers” used here does not seem appropriate as a medical term.

A: we change painkillers into anti-inflammatory agents.

Page 7 heroin

The expression “vial disease-induced nephropathy” does not seem common and should be explained. It is unclear what is responsible for the kidney biopsy findings in the second half; if it is a direct effect of Heroin, it should be clearly stated.

A: we added a new reference 71:  Jaffe JA, Kimmel PL. Chronic nephropathies of cocaine and heroin abuse: a critical review. Clin J Am Soc Nephrol. 2006 Jul;1(4):655-67. doi: 10.2215/CJN.00300106. Epub 2006 Jun 21. PMID: 17699270 to show the association between heroin-users’ viral diseases and nephropathy/kidney injury. 

Page 8 line34-36

Examples of lipophilic and high molecular weight toxins should be provided.

A: we added new reference 78: McIntyre CW, Fluck RJ, Freeman JG, Lambie SH. Characterization of treatment dose delivered by albumin dialysis in the treatment of acute renal failure associated with severe hepatic dysfunction. Clin Nephrol. 2002 Nov;58(5):376-83. doi: 10.5414/cnp58376. PMID: 12425489 to explain the lipophilic and high molecular weight toxins removal therapy.

Reviewer 2 Report

Comments and Suggestions for Authors

Abstract is incomplete.

‘Arterial hypertension’ instead od ‘hypertension’.

‘Prerenal’ instead of ‘pre-renal’.

Maybe better formulation: “Notably, urine discoloration can offer valuable insights into AKI etiology. In rhabdomyolysis, where muscle cell…”

I would suggest to try different formulation for this part of the sentence: “such as renal causes, such as rhabdomyolysis”.

The research analyzes topic of AKI. There are no novelties in the paper. It simply covers the review of AKI. The paper covers useful information but can be composed in a more coherent way. The order could be better. For example, epidemiology could be addressed after the definition. Symptoms and clinical image could be mentioned later in the text. The references seem appropriate.

Comments on the Quality of English Language

Additional language editing is recommended.

Author Response

Thank you for the great comment, and we revised it accordingly.

Comments and Suggestions for Authors

Abstract is incomplete.

A: we have re-written it as:

Acute kidney injury (AKI) is a commonly seen debilitated disease in clinical practice. In the emergency department, there are several aspects of etiologies causing AKI. We introduce the AKI definition, symptoms, causes, epidemiology, poisoning-related AKI, mechanism, etiology, and laboratory tests. AKI in the emergency department, etiologies, epidemiology and risk factors. We discuss the strategy in toxin, substance-related AKI such as Glafenin, antimicrobial agents, lithium, contrast media, snake venom, herbicide, ethylene glycol, synthetic cannabinoids, cocaine, heroin, and amphetamine. Then we reviewed the management of AKI and talked about prevention and outcome of AKI. This gives an overview of poisoning-related AKI.

‘Arterial hypertension’ instead od ‘hypertension’.

‘Prerenal’ instead of ‘pre-renal’.

A: We have revised them.

Maybe better formulation: “Notably, urine discoloration can offer valuable insights into AKI etiology. In rhabdomyolysis, where muscle cell…”

I would suggest to try different formulation for this part of the sentence: “such as renal causes, such as rhabdomyolysis”.

A: Thanks for the valuable suggestion, and we list variable factors of urine discoloration such as rhabdomyolysis, infection, drugs, or toxins, and they are somewhat clear-cut difficult to divide into renal causes and rhabdomyolysis-caused. That is, some are systemic (infection and drugs.)

The research analyzes topic of AKI. There are no novelties in the paper. It simply covers the review of AKI. The paper covers useful information but can be composed in a more coherent way. The order could be better. For example, epidemiology could be addressed after the definition.

A: Thank you, and we have already put the epidemiology (references 5 to 8) after the definition (references 1 to 4).

Symptoms and clinical image could be mentioned later in the text. The references seem appropriate.

A: thank you, we have put the symptoms (references 9 to 11) after the definition and epidemiology (references 1 to 8) already.

Comments on the Quality of English Language

Additional language editing is recommended.

A: We have completed the English editing service via https://www.mdpi.com/authors/english

Submission Date

13 July 2024

Date of this review

31 Jul 2024 23:50:29

Round 2

Reviewer 1 Report

Comments and Suggestions for Authors

The authors have generally corrected, though not fully, the issues raised.

I believe that this paper is worthy of acceptance.

Reviewer 2 Report

Comments and Suggestions for Authors

Just a smaller error: ”we also discuss its mechanism”

The article is nicely written, summarized and easily readable.

This topic is always of interest.